# Optimizing the Outcomes of Percutaneous Coronary Intervention in Patients with Chronic Kidney Disease

**DOI:** 10.3390/jcm11092380

**Published:** 2022-04-23

**Authors:** Alessandro Caracciolo, Renato Francesco Maria Scalise, Fabrizio Ceresa, Gianluca Bagnato, Antonio Giovanni Versace, Roberto Licordari, Silvia Perfetti, Francesca Lofrumento, Natasha Irrera, Domenico Santoro, Francesco Patanè, Gianluca Di Bella, Francesco Costa, Antonio Micari

**Affiliations:** 1Department of Clinical and Experimental Medicine, Policlinic “Gaetano Martino”, University of Messina, 98100 Messina, Italy; caracciolo.alessandro.ac@gmail.com (A.C.); rfm.scalise@gmail.com (R.F.M.S.); gianbagnato@gmail.com (G.B.); antonio.versace@polime.it (A.G.V.); robertolicordari@gmail.com (R.L.); silvia.perfetti@hotmail.it (S.P.); francesca.lofrumetno@studenti.unime.it (F.L.); natasha.irrera@unime.it (N.I.); domenico.santoro@unime.it (D.S.); gianluca.dibella@unime.it (G.D.B.); 2Department of Cardio-Thoraco-Vascular Surgery, Division of Cardiac Surgery, Papardo Hospital, 98158 Messina, Italy; ceresa77@hotmail.com (F.C.); f_patane@hotmail.it (F.P.); 3Department of Biomedical and Dental Sciences and Morphological and Functional Imaging, University of Messina, 98100 Messina, Italy

**Keywords:** chronic kidney disease, percutaneous coronary intervention, contrast-induced nephropathy

## Abstract

Percutaneous coronary intervention (PCI) is one of the most common procedures performed in medicine. However, its net benefit among patients with chronic kidney disease (CKD) is less well established than in the general population. The prevalence of patients suffering from both CAD and CKD is high, and is likely to increase in the coming years. Planning the adequate management of this group of patients is crucial to improve their outcome after PCI. This starts with proper preparation before the procedure, the use of all available means to reduce contrast during the procedure, and the implementation of modern strategies such as radial access and drug-eluting stents. At the end of the procedure, personalized antithrombotic therapy for the patient’s specific characteristics is advisable to account for the elevated ischemic and bleeding risk of these patients.

## 1. Introduction

Percutaneous coronary intervention (PCI) is one of the most common procedures performed in medicine [1]. PCI improves survival in acute coronary syndrome and helps to control anginal symptoms in chronic coronary disease [2,3]; however, its net benefit among patients with chronic kidney disease (CKD) is less well established. The use of contrast dye, arterial wall instrumentation, and the potential for microembolization are associated with potential renal harm which is amplified in patients with pre-existing CKD, potentially reducing the clinical benefit of PCI, especially in an elective setting.

The clinical impact of PCI in patients with stable CAD has been widely studied in recent years: in the ISCHEMIA trial [4], 5179 patients with stable coronary disease and moderate to severe inducible ischemia by imaging test were randomized to an immediate invasive strategy with coronary angiography or an initial approach with medical therapy alone. Over a median of 3.2 years, the primary outcome of cardiovascular death, myocardial infarction, resuscitated cardiac arrest, or hospitalization for unstable angina or heart failure was similar in the two treatment groups, confirming the modest impact of an initial invasive strategy in patients with stable angina and the potential for early related complications. This concept is even more important for patients with CKD, especially in its more severe forms. CKD and coronary artery disease (CAD) are strictly related, and are associated with a higher risk of thrombotic and bleeding complications [5]. In fact, lower values of glomerular filtration rate (GFR) below 60–75 mL/min/1.73 m^2^ are associated with a linear increase in CAD risk and a tripled risk of cardiovascular mortality when reaching GFR drops below 15 mL/min/1.73 m^2^ [6]. In addition, the impaired renal elimination of antithrombotic drugs exposes patients with CAD and PCI to a higher risk of bleeding complications. 

There is a paucity of data regarding the impact of PCI on patients with CKD, especially those in advanced stages or those treated with dialysis, who are often excluded from clinical trials. Recently, the ISCHEMIA CKD trial included 777 patients with advanced renal insufficiency (eGFR < 30 mL/min) in the context of the larger ISCHEMIA trial population. As observed in the main population, an early routine invasive strategy failed to reduce the incidence of death or myocardial infarction, and an excess of stroke, death or the initiation of dialysis was observed compared to the initial approach with medical therapy alone [7]. In addition, for CKD, no benefits of an early invasive strategy were evident with regard to angina-related health status [2]. 

Hence, the PCI benefit window for CKD patients is narrow. After careful patient selection, careful technological and organizational strategies should be implemented in order to allow a positive trade-off, balancing the risk of the procedure with the potential benefits. In this review, we will discuss strategies to minimize PCI’s potential for harm in CKD patients, in addition to the current evidence for pharmacological and device therapies in this domain. 

## 2. Contrast-Induced Nephropathy Prophylaxis Strategies

Contrast-induced nephropathy (CIN) is defined as the impairment of renal function, with either a relative 25% increase or an absolute 0.5 mg/dL increase within 48–72 h of intravenous contrast administration. CIN is associated with an increased risk of all-cause death and ischemic events, and should be thoroughly prevented [8]. Several pathophysiological mechanisms of CIN have been proposed. Contrast dye might exert direct renal toxicity mediated by free radicals and oxidative stress, or indirect toxicity through medullary hypoxia due to a vasodilation/vasoconstriction imbalance. 

The risk of CIN depends on patient’s characteristics and procedural variables. Patients with pre-existing renal impairment are exposed to the highest risk of CIN, in direct relation to serum creatinine level [9]. Diabetes mellitus is a mild risk factor, but in the presence of renal dysfunction it has a synergistic effect that exposes patients to a four-fold higher risk of CIN [10,11,12,13]. Advanced age, heart failure, haemodynamic instability, anaemia, dehydration, female sex, procedural bleeding, nephrotoxic drugs, and type and dose of contrast are additional risk factors that increase the risk of CIN, especially when renal impairment coexists [14,15,16,17,18,19,20]. There are no effective treatments for CIN, so prevention represents the most important strategy. The risk of CIN should be estimated in every patient considering clinical history and renal function. An estimated glomerular filtration rate (eGFR) lower than 60 mL/min is suggestive of a high risk for CIN [21]. In this setting, reducing the total volume of contrast administered is key to preventing CIN. A ratio of total contrast volume administered (in mL) to eGFR (in mL/min) higher than 3.7 exposes patients to a higher risk of CIN [17,18]. Several methods [15,17,22,23,24,25] have been proposed to identify high-risk patients, but there is no evidence that suggest their systematic use. Nevertheless, it is recommended that particular attention should be paid to the pre- and post-procedural clinical management of patients that present clinical characteristics associated with an increased risk of CIN [26]. Nephrotoxic drugs should be suspended before the procedure, and drugs that have an impact on renal function should be carefully evaluated for their benefit/risk ratio [21]. Studies on CIN prophylaxis have focused on three main strategies: fluid administration, pharmacological prevention, and renal replacement therapies.

### 2.1. Fluid Administration

Hydration represents the most important strategy for CIN prevention before and after PCI in patients with CKD. Fluids administration expands plasma volume, determines a downregulation of the renin–angiotensin–aldosterone system, reduces renal cortical vasoconstriction, dilutes contrast agents, and prevents tubular obstruction [27]. The European Society of Cardiology recommends (class I, level of evidence C) to administer isotonic saline to all patients with moderate to severe CKD (1 mL/kg/h or 0.5 mL/kg/h in patients with LVEF ≤ 35% or NYHA > 2) 12 h before and 24 h after the procedure [28]. Intravenous hydration has been found to be more effective in reducing CIN than oral hydration, although intravenous administration appears to be less manageable in acute patients and in day-hospital settings [29,30]. A meta-analysis of six trials showed no differences between intravenous and oral hydration in CIN reduction, suggesting that further adequately powered trials are needed [31]. Intravenous hydration with 0.9% isotonic saline was found to be more effective in CIN reduction when compared to other solutions [32]. Fluid administration in patients with renal impairment is often performed at a flow significantly lower than that assumed to give protection because of the concern of volume overload, especially in patients with left ventricle dysfunction [33]. Different combined strategies of hydration and diuretics have been tested under the assumption that a higher urine output relates to a greater contrast dilution and lower contrast toxicity. Loop diuretics showed a negative effect probably related to volume depletion and consequent vasoconstriction [34]. Interestingly, it has been demonstrated that increased diuresis, concurrently obtained with diuretics from fluid administration matched with urine output, not only prevents dehydration but also reduces CIN occurrence [35]. On the basis of this evidence, an automated hydration system was developed (Renal Guard System™, Renal Guard Solutions, Inc., Milford, MA, USA). The Renal Guard consists of a collection bag for the urine with a computed monitoring system and an intravenous infusion system. After an initial bolus of 250 mL isotonic saline and furosemide (0.25–0.5 mg/kg) that stimulates the diuresis, the saline solution is constantly infused at a volume corresponding to the volume of urine that flows into the collection bag, minimizing the risk of fluid depletion or overload. The Renal Guard System™ was tested for efficacy and safety in a clinical trial, and it was associated with a statistically significant reduction in CIN and the need for renal replacement therapy [36,37,38]. Interestingly, the comparison of the mean volume of fluids administered with the Renal Guard System™ and the typical hydration protocol (4000 mL vs. 1750 mL) in relation to timing (6 h vs. 24 h) emphasized the beneficial effect of hydration [37].

### 2.2. Pharmacological Prevention

Several pharmacological therapies have been evaluated for CIN prevention [39] in patients with CKD, producing mainly contrasting results. Among those tested, N-acetylcysteine, bicarbonate and statins represent the most promising molecules.

N-Acetylcysteine has been tested for CIN prevention on the basis of its antioxidant and vasodilator effects. A protective role of N-acetylcysteine was demonstrated in an initial study [40], but this was not confirmed in the subsequent clinical trials [41,42,43] that showed conflicting results, nor in the pooled data from the meta-analysis [44,45,46,47,48,49,50,51,52]. A multitude of elements may have contributed to the heterogeneity of these results, e.g., differences in patient selection, the definition of CIN, contrasting agent types, concomitant fluid administration, or different administration route [53]. Interesting, it was observed that the greater protective effect of N-acetylcysteine was in patients that received small amounts of contrast (<140 mL) [43]. Additional studies that tested higher cumulative doses of N-acetylcysteine compared with the most used protocol demonstrated a protective effect [54], suggesting a dose-dependent mechanism. A subsequent meta-analysis regarding the protective effect of high doses of N-acetylcysteine suggested that the most beneficial effect might be obtained in patients at high risk of CIN [55]. The beneficial effect of high doses of N-acetylcysteine on CIN prevention was confirmed in patients with STEMI [56], and it was also associated with a reduction in hospital deaths. Furthermore, in patients with myocardial infarction treated with intravenous N-acetylcysteine, a smaller size of infarcted area and protection of the left ventricular function were observed [57,58]. In acute ischemic settings, N-acetylcysteine antioxidant properties are potentially able to lessen oxidative stress related to reperfusion, and it has been demonstrated that this compound reduces platelet inhibition with a potential reduction in thrombotic burden [59]. A combination of hydration with sodium bicarbonate and N-acetylcysteine immediately before and up to 12 h after PCI reduced CIN occurrence compared to hydration with isotonic saline [60]. Conversely, in a trial of patients undergoing coronary angiography who were randomized to hydration with isotonic saline associated with oral N-acetylcysteine or hydration with bicarbonate, no differences emerged in terms of CIN prevention [48].

Alkalization therapy has been tested for CIN prevention in patients with CKD on the basis of a potential protective effect of renal tubular epithelial cells related to the reduction in renal tubules acidification with potential antioxidant effects [61], but conflicting results emerged. It was initially observed that hydration with bicarbonate was more effective in CIN reduction as compared to hydration with saline [62], but a subsequent study [63] and a meta-analysis did not confirm this data [64]. More recently, a randomized controlled trial showed no difference between intravenous sodium bicarbonate over intravenous isotonic saline or oral acetylcysteine over placebo for the prevention of CIN, need for dialysis, or death in patients with CKD undergoing coronary or non-coronary angiography [65].

Statins have been tested for CIN prevention in patients with CKD on the basis of their pleiotropic effect that includes anti-inflammatory activity and the improvement of endothelial function [66]. It was initially demonstrated that, in statin-naïve patients with acute coronary syndrome undergoing coronary angiography, a pre-treatment with rosuvastatin reduced CIN occurrence [67]. A meta-analysis of 124 trials and 28,240 patients comparing different strategies of CIN prevention in patients undergoing PCI demonstrated an important protective effect of premedication with statins [68] that, in accordance with another meta-analysis, was found to be independent of the hydration protocol. On the basis of these results, the European Society of Cardiology recommends (class IIa, level of evidence C) high-dose statins in statin-naïve patients.

Proton pump inhibitors (PPIs) have been demonstrated to reduce the rate of recurrent gastrointestinal bleeding in high-risk patients receiving aspirin [69]. Prior observational studies suggested a possible increased risk of cardiovascular ischemic events when PPI therapy was administered concomitantly with clopidogrel [70]. However, randomized trials did not support such concerns [71,72]. ESC guidelines endorse the routine association of PPI during DAPT treatment with class IB recommendations [73].

### 2.3. Renal Replacement Therapies

Haemodialysis and hemofiltration have been proposed for CIN prevention in patients with CKD because of their effectiveness in removing contrast agents from circulation. Several studies evaluating haemodialysis immediately after an angiographic procedure [74,75,76,77] failed to demonstrate a beneficial effect. It has been proposed that the lack of clinical benefit of haemodialysis could be related to the nephrotoxicity of the procedure that determines a pro-inflammatory, pro-coagulative, hypotensive and hypovolemic effect [78]. Hemofiltration, as compared to haemodialysis, is often found to be more manageable because fluid and solute removal are performed with better volume control and more haemodynamic stability. Indeed, a randomised study demonstrated that hemofiltration reduces CIN occurrence and 1-year mortality in patients with severe CKD. A subsequent study [79] compared two different protocols of hemofiltration in patients with severe CKD; one group was treated with hemofiltration for 6 h before and for 18/24 h after the procedure, while the other group was treated for 18/24 h after the procedure. An important reduction in CIN occurrence was observed in the group treated with hemofiltration before and after the procedure. On the basis of these results, the European Society of Cardiology recommends (class IIb, level of evidence B) hemofiltration for 6 h before and for 24 h after a given procedure for patients with severe CKD undergoing complex PCI, while prophylactic haemodialysis is not recommended (class III, level of evidence B) [28].

## 3. Transradial Artery Access

Transradial access (TRA) for percutaneous coronary angiography and intervention has become the default route over the transfemoral approach (TFA) on the basis of several advantages [80]. Among these advantages is an important reduction in renal complications [81,82,83,84]. Different mechanisms have been implicated to explain the nephroprotective effect of TRA over TFA, including a lower incidence of major bleeding [20,85], embolization [86,87] and hypotension, but also a lower use of contrast agents. Several studies have evaluated the benefit of TRA on renal outcome with heterogeneous results [88,89,90]. In the pivotal MATRIX-Access randomized trial, TRA was demonstrated to reduce major bleeding and all-cause mortality compared to TFA [91]. A prespecified sub-analysis of this study revealed a reduction in acute kidney injury occurrence in the TRA cohort compared to the TFA cohort [92]. Interestingly, a subsequent multistate and competing risk model analysis suggested that the reduction in mortality was mainly mediated by the reduction in acute kidney injury [93]. Finally, a large meta-analysis of 14 studies and 46,816 patients confirmed that TRA was associated with a lower occurrence of acute kidney injury after coronary angiography or PCI [94], compared to TFA.

## 4. Contrast Dye Reduction for Coronary Angiography and PCI

In recent years, new procedural approaches have been introduced to reduce contrast administration with the aim of performing ultra-low contrast angiography and virtually zero-contrast PCI. The implementation of these techniques could reduce the need for contrast during the procedure to a minimum. For example, catheter engagement in the coronary ostia could be performed without contrast by focusing on calcium distribution using a high frame rate [95]. Moreover, correct cannulation could be confirmed without contrast by injecting 10–20 mL of isotonic saline and observing temporal changes in the electrocardiogram (i.e., T wave or ST segment modification) [96] or, alternatively, by cautiously advancing a coronary guidewire [97]. A total of 15 mL of contrast is usually sufficient to perform a reliable coronary angiography [98] by injecting 2–3 mL of contrast to visualize the left coronary artery and 2 mL for the right coronary artery. Contrast medium should be removed from the catheter prior to every drug administration or catheter exchange, and contrast must be refilled before subsequent angiography. When clear angiographic images are available, PCI without contrast may be attempted (during the same session or in a staged procedure) using different techniques, intravascular imaging, and functional tests. Large guiding catheters (usually 7 Fr) are preferable as they give stable support and accommodate multiple guidewires, stents and IVUS probes. Multiple guidewires should trace the course of the vessels shown on the reference angiography, and represent a map used to track major reference points during the procedure (e.g., ostia, bifurcations etc.). Once the correct position of the guide wire has been verified, an IVUS evaluation can be performed. IVUS is able to accurately define plaque burden and reference vessel dimensions, allowing the selection of the optimal stent size [97,98]. Several landmarks such as calcification, ribs, surgical clips, the catheter, or guidewires are useful to gain the correct position for stent placement. After stent implantation, correct stent expansion and possible dissections can be assessed by IVUS. IVUS co-registration to merge angiography and intravascular probe position could further increase PCI accuracy. The physiological evaluation of coronary plaque can be used to guide the procedure and confirm the effectiveness of the intervention [99]. In non-complex anatomy, rotational atherectomy can be performed without contrast, because calcifications of the wall vessel usually mark the location and the extent of the lesion [100]. If vessel perforation, distal embolization, or other complications that cannot be ruled out by IVUS or functional tests are suspected, a small contrast injection can elucidate the problem.

## 5. Revascularization Strategy

Coronary atherosclerosis in patients with CKD is typically associated with a higher burden of calcification, and more frequently involves the left, main, or three vessels resulting in high lesion complexity [101,102,103]. A correlation between the severity of renal impairment and coronary lesion complexity expressed by an inverse relationship between the eGFR and the SYNTAX (SYNergy between PCI with TAXUS™ and Cardiac Surgery) Score has been demonstrated [104]. In the clinical setting of ACS, CKD affects nearly 30–40% of patients and it has been demonstrated to be an independent predictor of death and MACCE with a correlation between the severity of CKD and the event rate [105,106,107]. Despite this, patients with ACS and CKD less frequently receive optimal medical treatment and early invasive strategies [107,108]. ACS diagnosis in patients with CKD may be delayed due to atypical presentation without chest pain, ECG abnormalities, or mild elevations in cardiac necrosis markers. Furthermore, CKD (particularly end-stage renal disease) has been adopted as an exclusion criteria for large ACS clinical trials, so the efficacy/safety profile of different treatments remain uninvestigated against the disease [109]. Interestingly, CKD patients with ACS undergoing PCI have been evaluated with three-vessel grayscale and virtual histology intravascular ultrasound (IVUS) imaging [110]. Longer atherosclerotic lesions with augmented necrotic core-to-fibrous cap ratios, higher plaque burdens, greater luminal inclusions, and the coexistence of these elements of complexity have been proposed as evidence for the increased risk of periprocedural complications [111,112].

The choice of the best strategy of revascularization is critical to improve the patient’s prognosis. Several studies have shown that drug-eluting stents (DES) are superior to bare-metal stents (BMS) in reducing MACE at a distance [113], irrespective of clinical and procedural characteristics [114]. Similar results have been obtained in the CKD population. Crimi et al. compared the impact of BMS vs. DES (paclitaxel-PES; zotarolimus-ZES-S; everolimus-EES-eluting stent) implantation in patients with CKD (GFR < 60 mL/min/1.73 m^2^) in a post hoc analysis of the PRODIGY trial. A total of 2003 patients with stable or unstable CAD were randomized 1:1:1:1:1 to BMS-EES-PES-ZES. The study showed that CKD at baseline was associated with a two-fold higher risk of stent thrombosis (ST) and MACE, and EES halved ST risk at 2 years after PCI in CKD patients compared with BMS and PES [115]. Bangalore et al. evaluated, in an observational study, the impact of different revascularization strategies in CKD patients with multivessel disease, comparing outcomes for CABG and PCI with everolimus-eluting stents. This study showed that, in patients with CKD, CABG was associated with a higher short-term risk of death, stroke, and repeat revascularization, while PCI was associated with a higher long-term risk of repeat revascularization and myocardial infarction. In the subgroup of patients on dialysis, the results favored CABG over PCI [116].

## 6. Secondary Prevention Antithrombotic Drug

To improve the prognosis of patients with CKD, the correct management of antithrombotic therapy as a secondary prevention factor is of great importance. Despite huge improvements in antithrombotic therapy for secondary prevention, the prevalence of CKD patients within the randomized populations of pivotal clinical studies has been low. In the PLATO trial, comparing ticagrelor vs. clopidogrel in patients with ACS, the proportion of CKD patients was only 21.3%, while dialysis was a study exclusion criteria. Similarly, in the TRITON TIMI 38 trial testing prasugrel vs. clopidogrel in ACS patients, the proportion of CKD patients was only 15.1%, and patients with CKD stage >4 were rare [117,118].

CKD patients carry both a higher ischemic and bleeding risk [119]. The coagulation cascade is imbalanced towards more thrombotic activity: the concentration of prothrombotic factors including fibrinogen, tissue factor and higher inflammatory milieu increase the risk of thrombotic complications [120]. The endothelial injuries associated with CKD also favor the loss of antithrombotic properties [121]. On the other hand, CKD might impair α-granule release and prostaglandin metabolism, impairing platelet aggregation [122]. Circulating fibrinogen fragments interfere by competitive binding to the glycoprotein IIb/IIIa receptor, resulting in decreased adhesion and aggregation and increased bleeding liability [123]. In addition, altered drug metabolism increases plasma concentration and the risk of antithrombotic overdosing in CKD patients.

Multiple studies have shown a high platelet reactivity in patients with CKD [124,125]. In the ADAPT-DES registry, authors compared platelet function in patients with and without CKD, demonstrating that those with CKD had higher platelet reactivity with a linear relationship to the renal function of platelet function testing [126]. Similarly, Angiolillo et al. observed that diabetic patients with CKD had markedly elevated platelet reactivity with a reduced response to the active metabolite of clopidogrel, suggesting altered P2Y_12_-mediated signaling [127]. High platelet reactivity with reduced responsiveness to clopidogrel (the best-studied P2Y12I) resulted in an increase in MACE and ACS in patients treated with PCI [128,129]. In the CREDO (Clopidogrel for the Reduction of Events During Observation) trial, the CKD group patients did not show the same benefit in terms of reduction in death, myocardial infarction, or stroke, with respect to the placebo of the non-CKD group [130]. The evidence for potent P2Y12i is limited in CKD patients. A small single-center study enrolling non ST-elevation myocardial infarction patients with CKD demonstrated significantly lower P2Y12 reaction unit (PRU) values in the group treated with ticagrelor versus the clopidogrel group [131]. In another small study, prasugrel demonstrated no difference in the pharmacokinetics and pharmacodynamics of subjects with CKD in terms of platelet inhibition contrary to clopidogrel [132,133]. A comparison of three P2Y12Is (ticagrelor, prasugrel and clopidogrel) in CKD patients with ACS undergoing PCI were available in the subgroup analysis of the two RCTs (TRITON TIMI 38 and PLATO studies). The TRITON TIMI 38 subgroup analysis included 1490 patients with a creatinine clearance < 60 mL/min. In this group, the benefit of prasugrel treatment compared with clopidogrel was similar to that of the overall population without significant interaction between the treatment groups and the CKD group [117]. The CKD sub-group of the PLATO trial was composed of 3237 patients who were followed over a mean of 9 months. No interaction was noted between the CKD and treatment groups; this data suggests a similar net benefit ratio in patients with CKD compared to the normal population (*p* = 0.13). On the other hand, all-cause mortality was lower in the ticagrelor group, with a 36% reduction (3.9 versus 5%; *p* = 0.01). The results were not significant for non-CKD patients [134]. The SWEDEHEART registry compared two P2Y12Is (ticagrelor and clopidogrel) in patients undergoing PCI for ACS and suffering with CKD. The registry defined two groups: moderate CKD (30–60 mL/min) and severe CKD (<30 mL/min). In the moderate CKD group, lower rates of death, myocardial infarction, and stroke at the 1-year follow-up were registered in the ticagrelor group compared with the clopidogrel group (adjusted HR, 0.82; 95% CI, 0.7–0.97). No benefit was observed for the patients with severe CKD (adjusted HR, 0.95; 95% CI, 0.69–1.29). Bleeding with the need for hospitalization was similar between the two groups in the moderate CKD setting (OR, 1.13; 95% CI, 0.84–1.51), while a trend towards higher bleeding rates was recorded in patients in the severe CKD group treated with ticagrelor (adjusted HR, 1.79; 95% CI, 1.00–3.21) [135]. The TRILOGY ACS trial (Targeted Platelet Inhibition to Clarify the Optimal Strategy to Medically Manage Acute Coronary Syndromes) was a randomized study comparing prasugrel and clopidogrel over 30 months in combination with aspirin in an ACS medically managed setting. A prespecified subgroup analysis showed that patients with moderate or severe CKD had an excessive risk of ischemic and bleeding events, and there were no differences between prasugrel and clopidogrel in terms of outcome in these subgroups [136].

Dual antiplatelet therapy duration should be selected on a single-patient basis, taking into consideration clinical characteristics [137,138] and CKD [139]. In a post hoc analysis of the PRODIGY trial, CKD did not appear as a treatment modifier for DAPT duration with respect to ischemic events, and longer DAPT was associated with excessive bleeding in moderate and severe CKD [140]. On the other hand, CKD is included in several risk scores as a criterium for high ischemic and bleeding risk. In a sub-analysis of the PRECISE-DAPT population, shorter-term DAPT was associated with improved outcomes for patients that were considered at higher ischemic and bleeding risk, supporting the concept that when both ischemic and bleeding risk are high, bleeding risk prevention with shorter-term DAPT is preferred [141]. Concomitant treatment with proton pump inhibitors while patients are on DAPT magnifies the benefit of antithrombotic therapy by limiting gastrointestinal bleeding [72,142], and should be maintained throughout.

## 7. VKA/NOAC for Atrial Fibrillation (AF) in Patients with CKD

Atrial fibrillation and CKD have a high prevalence in the adult population, and frequently co-exist in the same patient. Patients with AF and CKD have both thromboembolic and hemorrhagic risk factors that significantly contribute to elevated mortality and morbidity in this population [143]. This framework particularly concerns the metabolism of the four currently available NOAC compounds that are all partially excreted by the kidneys: 80% of dabigatran is eliminated thorough renal clearance, while 50%, 35%, and 27% of edoxaban, rivaroxaban, and apixaban are eliminated this way, respectively. To date, no RCT has investigated the clinical role of VKA for thromboprophylaxis in AF patients with severe or end-stage kidney disease and, unfortunately, the main trials with NOACs excluded patients with an eGFR lower than 30 mL/min. Beyond that, on the basis of pharmacokinetic analysis, rivaroxaban, apixaban, and edoxaban (but not dabigatran) are approved in Europe for patients with severe CKD eGFR: 15–29 mL/min) with a reduced dose regimen.

Patients with AF undergoing PCI are an increasingly large group [144] and, in this specific clinical setting, CKD is a crucial variable to consider to properly tailor antithrombotic treatment, as this a major criterion for higher bleeding risk [145]. The AGUSTUS trial, with a 2 × 2 design, evaluated the safety of apixaban AVK, aspirin, and placebo in patients with ACS and/or undergoing PCI. Within the study population, eGFR was >80 mL/min in 30% of patients, >50–80mL/min in 52% of patients, and 30–50 mL/min for 19% of patients. Patients treated with apixaban—compared with VKA—had a lower rate of death, hospitalization and bleeding, independent of renal function [146]. Data from the REDUAL-PCI study are aligned with these results. Patients with AF who had undergone PCI were assigned to triple therapy (VKA, aspirin, and clopidogrel or ticagrelor) or dual therapy with dabigatran (110 mg or 150 mg) and clopidogrel or ticagrelor. Dual therapy with dabigatran 110 mg, compared with VKA, reduced the risk of major bleeding events or clinically relevant non-major bleeding events irrespective of eGFR class (*p* for interaction = 0.19). Likewise, dual therapy with dabigatran 150 mg reduced the risk of major bleeding events or clinically relevant non-major bleeding events irrespective of eGFR class, compared with VKA. No significant differences in the prevention of thromboembolic events or unplanned revascularization emerged between dual therapy with dabigatran 110 mg or triple therapy, irrespective of eGFR class. Dual therapy with dabigatran 150 mg, compared with triple therapy, had a similar risk for thromboembolic events or unplanned revascularization in patients with eGFR from 30 to <80 mL/min, and a lower risk with eGFR ≥ 80 mL/min (*p* for interaction = 0.02) [147].

## 8. Optimal Medical Therapy for CKD

### 8.1. Hypertension Treatment

Hypertension is the second most important cause of CKD and an independent risk factor for cardiovascular events [148]. The evidence suggests that blood pressure (BP) targets should be lowered to <140/90 mmHg with the aim of moving towards 130/80 mmHg [149,150]. The SPRINT trial demonstrated that a more ambitious systolic BP target < 120 mmHg reduces CV events and all-cause mortality compared to a target of <140 mmHg [151]. CKD guidelines suggest a combination of renin angiotensin system (RAS) blockers with calcium channel blockers (CCBs) as a first-choice therapy [152]. Other therapeutic agents, such as beta-blockers, spironolactone, diuretics (amiloride, thiazide, thiazide-like diuretics or loop diuretics), and alpha-blockers could be added as second-line therapy options [153]. Many therapeutic agents should be considered and monitored carefully for their impact on renal function and potassium levels, especially in patients with eGFR < 45 mL/min/1.73 m^2^ and serum potassium levels > 5.0 mmol/L.

### 8.2. Lipid Control

According to ESC guidelines, patients with advanced CKD are considered to be at high or very high risk of cardiovascular disease, with LDL targets of 70 mg/dL and 55 mg/dL, respectively [154]. The KDIGO organization developed practice guidelines for the management of dyslipidemia in CKD patients in which the use of statins or a statin/ezetimibe combination was recommended in non-dialysis patients with end-stage CKD. For dialysis patients who are already on lipid-lowering agents at the time of dialysis initiation, the continuation of these drugs is recommended, especially in cases where there is evidence of atherosclerotic cardiovascular disease [155].

The metabolisms of statins are mainly explained by the liver (and minimally by the kidneys), so dose adjustment with CKD and hemodialysis is not necessary, with the exception of hydrophilic statins such as pravastatin and rosuvastatin. This particular kind of statin has a higher risk of myopathy and rhabdomyolysis in CKD patients [156]. For rosuvastatin, a dose adjustment in non-dialysis severe renal impairment with a starting dose of 5 mg, once daily, and a maximal recommended dose of 10 mg, once daily, is suggested.

## 9. Conclusions

The prevalence of patients suffering from both CAD and CKD is high and is likely to increase in the coming years. Planning adequate management of this group of patients is crucial to improve their outcome after PCI. This starts with proper preparation before the procedure, the use of all available means to reduce contrast use during the procedure, and the implementation of modern strategies such as radial access and drug-eluting stents. At the end of the procedure, a personalized antithrombotic therapy plan based on patient’s characteristics is advisable in light of their elevated ischemic and bleeding risk (Figure 1).

## Figures and Tables

**Figure 1 jcm-11-02380-f001:**
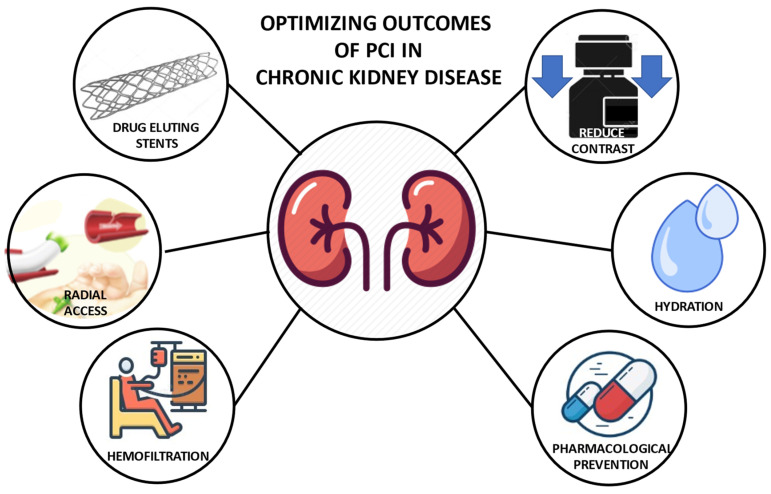
Strategies to optimize PCI outcomes in patients with chronic kidney disease.

## Data Availability

The data presented in this study are available in manuscript.

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
