# Peer review of "Optimizing the Outcomes of Percutaneous Coronary Intervention in Patients with Chronic Kidney Disease"

_jcm, 2022, doi:10.3390/jcm11092380_

Round 1
Reviewer 1 Report
The article is well written, informative and clear. The authors should be congratulated.
Please review this phrase "Creatinine Value (Finn, n.d.)10."
Does "ST risk" mean risk of ST elevation myocardial infarction?
Author Response
We thank the Reviewer for the valuable comments. We have revised the manuscript as suggested.
Reviewer 2 Report
The authors have described the current status of optimizing the outcomes of PCI in patients with chronic kidney disease.
I have the following comments
- The revascularization strategy can be more elaborated.
- A paragraph on OMT other than DAPT may be added
- PCI in ACS with CKD is also worth mentioning
- Left Main subset in CKD can also be incorporated
- Role of NOAC/VKA in AF, CKD and PCI
- ESC gives class I status for PPI with DAPT
Author Response
We thank Reviewer 2 for the valuable comments and insights. We have now revised the manuscript accordingly and found that the whole article is greatly improved.
Here following the point-by-point response to the Reviewers comments
- The revascularization strategy can be more elaborated.
We have now expanded this section as suggested adding the following text: "Coronary atherosclerosis in patients with CKD is typically associated with an higher burden of calcification and it frequently involves left main or three vessel resulting in more complex lesion 106–108. It has been demonstrated a correlation between the severity of renal impairment and the coronary lesion complexity that is expressed by an inverse relationship between the eGFR and the SYNTAX (SYNergy between PCI with TAXUS™ and Cardiac Surgery) Score 109 . In the clinical setting of ACS, CKD affects nearly 30-40% of patients and it has been demonstrated to be an independent predictor of death and MACCE with a correlation between severity of CKD and events rate 110,111,112.Despite this, patients with ACS and CKD less frequently receive optimal medical treatment and early invasive strategy 112,113 ACS diagnosis in patients with CKD may be delayed due to atypical presentation without chest pain, ECG abnormalities and mild elevation in cardiac necrosis marker. Furthermore, CKD (particularly end-stage renal disease), has been adopted as an exclusion criteria for large ACS clinical trial so the efficacy of different treatment strategies remain uninvestigated 114. Interestingly, CKD patients with ACS undergoing PCI have been evaluated with a 3-vessel grayscale and virtual histology intravascular ultrasound (IVUS) imaging 115 with the evidence of longer atherosclerotic lesions with an augmented necrotic core-to-fibrous cap ratio, higher plaque burden and greater luminal inclusion and, the coexistence of these elements of complexity, has been proposed to justify the increased risk of periprocedural complication 116,117" - A paragraph on OMT other than DAPT may be added
As suggested we have now added a new chapter regarding "Optimal Medical Therapy in CKD". This tackle both optimal hypertension and lipid control treatment in CKD patients undergoing PCI. - PCI in ACS with CKD is also worth mentioning
We have now expanded this elements in the section for revascularization as pointed in comment 1 to Reviewer 2. - Left Main subset in CKD can also be incorporated
We have now brielfy expanded on this matter in the revascularization section. Yet, we preferred not to include a dedicated chapter on LM or complex PCI in CKD for the sake of space as the main focus of the document is the optimization of outcomes through pre,peri and postprocedural treatment rather than focus on technical aspects of the procedure. - Role of NOAC/VKA in AF, CKD and PCI
We again thank Reviewer 2 for this comment. We added now a dedicated novel chapter expanding on the OAC therapy in CKD patients. - ESC gives class I status for PPI with DAPT
We have now added as suggested a paragraph regarding PPI in section 2.2 pharmacological prevention as follows: "Proton pump inhibitor (PPI) demonstrated to reduce the rate of recurrent gastrointestinal bleeding in high-risk patients receiving aspirin 71. Prior observational studies suggested a possible increase in risk of cardiovascular ischemic events when PPI therapy was administered concomitantly with clopidogrel 72. Yet randomized trials did not support such concerns 73,74 . ESC guidelines endorse routine association of PPI during DAPT treatment with a class IB recommendation 75"